# Pharmacological Treatment of Interstitial Lung Diseases: A Novel Landscape for Inhaled Agents

**DOI:** 10.3390/pharmaceutics16111391

**Published:** 2024-10-29

**Authors:** Vito D’Agnano, Fabio Perrotta, Ramona Fomez, Valerio Maria Carrozzo, Angela Schiattarella, Stefano Sanduzzi Zamparelli, Raffaella Pagliaro, Andrea Bianco, Domenica Francesca Mariniello

**Affiliations:** 1Department of Translational Medical Sciences, University of Campania L. Vanvitelli, 80131 Naples, Italy; vito.dagnano@studenti.unicampania.it (V.D.); ramona.fomez@studenti.unicampania.it (R.F.); valeriomaria.carrozzo@studenti.unicampania.it (V.M.C.); angela.schiattarella@studenti.unicampania.it (A.S.); raffaella.pagliaro@studenti.unicampania.it (R.P.); andrea.bianco@unicampania.it (A.B.); 2Unit of Respiratory Medicine “L. Vanvitelli”, A.O. dei Colli, Monaldi Hospital, 80131 Naples, Italy; 3Division of Pneumology, A. Cardarelli Hospital, 80131 Naples, Italy; stefano.sanduzzizamparelli@aocardarelli.it; 4UOC Pneumotisiologia Federico II, A.O.R.N. Monaldi-Cotugno-CTO Piazzale Ettore Ruggieri, 80131 Naples, Italy; domenica.mariniello@ospedalideicolli.it

**Keywords:** idiopathic pulmonary fibrosis, interstitial lung diseases, inhaled agents, antifibrotics, bronchodilators, corticosteroids, antioxidants

## Abstract

Interstitial lung diseases (ILDs) encompass a heterogeneous group of over 200 disorders that require individualized treatment. Antifibrotic agents, such as nintedanib and pirfenidone, have remarkably revolutionized the treatment landscape of patients with idiopathic pulmonary fibrosis (IPF). Moreover, the approval of nintedanib has also expanded the therapeutic options for patients with progressive pulmonary fibrosis other than IPF. However, despite recent advances, current therapeutic strategies based on antifibrotic agents and/or immunomodulation are associated with non-negligible side effects. Therefore, several studies have explored the inhalation route aiming to spread higher local concentrations while limiting systemic toxicity. In this review, we examined the currently available literature about preclinical and clinical studies testing the efficacy and safety of inhalation-based antifibrotics, immunomodulatory agents, antioxidants, mucolytics, bronchodilators, and vasodilator agents in ILDs.

## 1. Introduction

The term interstitial lung diseases (ILDs) refers to an extremely heterogeneous spectrum of about 200 diffuse pulmonary parenchymal disorders that share, to different extents, clinical, functional, radiological, and pathological similarities [1,2]. The pathological involvement of the interstitial space, bound by alveolar epithelium and pulmonary endothelium, represents the hallmark of ILDs [3]. However, pathological processes may extend beyond the “interstitium”, involving both the alveolar and vascular compartments. Inflammation and fibrosis—or the various combinations of both pathological mechanisms—characterize most of the ILDs [4]. Idiopathic pulmonary fibrosis (IPF) represents the prototype amongst the fibrotic progressive idiopathic interstitial pneumonias (IIPs). Likewise, other forms of IIPs as well as secondary ILDs, including connective tissue disease–ILDs (CTD-ILDs) and fibrotic sarcoidosis, are well recognized [5]. In addition, ILDs may coexist concurrently with other chronic pulmonary diseases, including emphysema, chronic obstructive pulmonary diseases, and pulmonary hypertension, which negatively impact outcomes [6,7]. Although the treatment landscape of ILDs has remained unchanged for years, the approval of novel antifibrotic medications has emerged as a real game changer for patients affected by IPF and other progressive fibrosing ILDs (PF-ILDs). Pirfenidone, an antifibrotic and anti-inflammatory agent capable of inhibiting fibroblast proliferation and collagen deposition by targeting the TGF-β pathway, has gained approval for the treatment of IPF with a recommended full oral dosage of 2403 mg/day divided into three doses [8]. Nintedanib is an intracellular multitarget tyrosine kinase inhibitor that is licensed not only for the treatment of IPF but also for other conditions. Nintedanib has also gained approval for PF-ILDs other than IPF as well as for systemic sclerosis–ILD (SSc-ILD) [9,10]. The recommended full oral dose is 150 mg, twice per day, or 100 mg twice per day in patients who do not tolerate the full dose or with mild hepatic impairment [11,12]. Regarding IPF treatment, there is no evidence or recommendation in the current guidelines about antifibrotic treatment preference. The decision should instead be the result of a tailored approach in which clinician–patient cooperation is crucial in reducing adverse effects and optimizing patient compliance. However, despite efforts, the burden of adverse effects often remains severe. These, usually, include dyspepsia, diarrhea, nausea, liver function abnormalities, and extra-digestive adverse effects [13]. The intent of this paper is to provide a broad perspective of the current state of the art and the development of inhalation-based agents in the treatment of ILDs that would offer the opportunity to tackle airway dysfunction, limit fibrogenetic processes, and improve endothelial dysfunction and pulmonary vasculature remodeling with a potentially more favorable safety profile.

## 2. Therapeutic Aerosols

Since antiquity, therapy with aerosols—a peculiar system of either solid particles or liquid droplets suspended in air—has represented one of the underpinning pillars of respiratory medicine and is based on one essential assumption: an inhaled medication can be directly delivered into the lungs [14,15,16,17,18]. This ideally implies that a smaller amount of the drug is needed, a more rapid onset is possible, and fewer systemic adverse effects occur compared to oral and/or parenteral administration. Three main mechanisms—inertial impaction, gravitational sedimentation, and Brownian diffusion—are recognized to dominate the quantity of aerosols that eventually will deposit into the airways [18]. Inertial impaction describes how particles with larger aerodynamic equivalent diameters deposit into the upper regions of the respiratory tract [19]. The aerodynamic equivalent diameter is defined as the diameter of a unit density sphere with a density of 1000 kg/m^3^ (1 g/cm^3^) and the same terminal settling velocity (Vt) of a given particle, where V_t_ is directly proportional to the square of the particle’s diameter as well as to the difference between the particle and its air density [19].

Deposition of particles with a diameter between 0.5 μm and 5 μm into the lung periphery is primarily governed by gravitational sedimentation. As a time-dependent mechanism, it is critically influenced by the patient’s respiratory pattern. A breath hold before exhalation may increase the likelihood of aerosol sedimentation [20]. Lastly, very small particles, with a diameter of less than 0.2 μm, primarily deposit via Brownian diffusion or are exhaled. Contrary to sedimentation, deposition via diffusion or Brownian motion increases with a decrease in particle size [19]. However, the deposition fraction—defined as the fraction of particles that are deposited and not exhaled—also depends on unpredictable patient factors, including airway anatomy, the rate of gas flow where aerosols are conveyed, and breathing patterns. In addition, mucus cilia and alveolar macrophages are recognized factors affecting aerosol deposition in the lung. Whilst mucus—mainly composed of human MUC5B mucin (MUC5B) and MUC5AC mucin (MUC5AC)—represents the primary clearance mechanism in the bronchial and bronchiolar regions, phagocytosis mediated by alveolar macrophages is responsible for aerosol clearance in peripheral air spaces [9,10].

### 2.1. Therapeutic Aerosols in Patients with ILDs: Aerosol Drug Delivery

Originally, the TOPICAL study documented the proof of concept that inhaled salbutamol can successfully reach the peripheral regions of the lung in a small cohort of patients with IPF [21]. Since then, efforts have been invested to apply aerosol therapy technologies to the treatment of ILDs. Moreover, concurrent lung diseases, such as COPD and asthma, for which inhalers are recommended, often coexist.

Regarding ILD treatment, several advantages of inhaled medications can be recognized. As stated above, smaller amounts of drugs may be used, and the safety profile improved. Currently, therapeutic aerosols can be generated through different mechanisms.

#### 2.1.1. Dry Powder Inhalers (DPIs)

Breath-activated, carrier-based, dry powder inhalers (DPIs) aerosolize micronized drug particles into airways [22]. The forced inhalation of a patient against the internal device resistance generates internal turbulence leading to drug particle separation from the carriers—usually lactose—which remain in the mouth whilst the drug is properly conveyed into the airways. A crucial feature affecting drug delivery through DPIs is the turbulence produced inside the device which depends on inhaler intrinsic resistance (R), the flow rate (Q), and the drop pressure, measured as peak inspiratory pressure (PIP); other host-related factors including airflow limitation, disease activity, and aging may finally influence drug delivery and pulmonary deposition [23,24]. Tyvaso DPIs represent a drug–device combination therapy that is currently approved for the treatment of pulmonary arterial hypertension (PAH) and ILD-PAH. Contrary to other DPIs, Tyvaso is not a carrier-based DPI. It, in fact, is based on the combination of a drug and an inert excipient, fumaryl diketopiperazine (FDKP) [25]. Its device (Dreamboat ^®^) allows for drug dispersion in less than 0.5 s with less than 500 mL of inhaled air volume. A user-generated inhalation pressure in the order of 2–4 kPa is required for prompting low flow rates around 20 L/min (16–22 L/min) through the DPI. The device permits the delivery of 16, 32, 48, or 64 μg doses of treprostinil in one breath [25].

#### 2.1.2. Pressurized Metered-Dose Inhalers (pMDIs)

Pressurized metered-dose inhalers (pMDIs) have represented the device of choice for decades for both the treatment of chronic respiratory diseases, including asthma and COPD, as well as for delivering inhalation agents in mechanically ventilated patients [26]. Three basic elements constitute a pMDI: a container where the drug is suspended along with a propellant gas—usually hydrofluoroalkane; a metered valve for dispensing the accurate dose of the drug; and a plastic mouthpiece. A pressure on the top of the metal container is requested to activate the pMDI. A more difficult hand–breath coordination represents a well-recognized drawback of pMDIs and a significant proportion of patients have reported experiencing difficulties in the inhalator technique [27]. Access to demo pMDI training with clinicians, as well as spacers, is thought to improve patient compliance to device and drug lung deposition, respectively. However, a paucity of evidence remains and trials are needed [27,28,29].

#### 2.1.3. Soft Mist Inhalers (SMIs)

Soft mist inhalers (SMIs) represent a group of low-velocity inhalation-generating devices characterized by high drug lung deposition (with up to > 50% of the dose) and capable of reducing some of the disadvantages of both pMDIs and DPIs, such as the necessity of propellant and difficulties of dry powder formulations, respectively. SMIs are also capable of aerosolizing a higher fraction of fine particles as well as lipid nanoparticle formulations (e.g., rhDNAse, mRNAs) [30]. From a technical perspective, SMIs are characterized by a remarkable reduction in spray velocity coupled with a longer plume spray duration, leading to a lower inertial impaction and greater dose reaching the lung. Despite less patient–inhaler coordination being needed, the inhalation flow rate may represent a patient-dependent key factor with relevant effects on final lung drug deposition [31].

#### 2.1.4. Nebulizers

Creating particles by shaking a liquid medication, nebulizers—jet, ultrasonic, and mesh nebulizers—are another well-established aerosol delivery method. Since they require modest patient cooperation and education, jet nebulizers (JNs) have represented the mainstay aerosol technique in emergency settings, in any disease severity, in older populations with cognitive impairments and children [32,33]. However, disadvantages are also recognized for JNs. They are bulky, require long treatment times, and provide inaccurate amounts of drugs to the airways; generally, doses fewer than 10% of the total are deposited into distal airways and the lungs [34].

Conversely, ultrasonic nebulizers generate ultrasonic waves via a piezo element rather than a bulky compressor. This leads to the formation and collapse of little bubbles, a phenomenon called cavitation. Bubble implosion, finally, produces waves that break down the solution at the solution. The obtained particles are bigger in size but more uniform compared to JNs [35].

#### 2.1.5. Vibrating Mesh or Membrane Nebulizers (VMNs)

Vibrating mesh or membrane nebulizers (VMNs) represent a novel alternative to JNs, overcoming some of their known limitations. In VMNs, the energy of ultrasonic waves driven by a piezo-electric element is used to vibrate a mesh, forcing the liquid drug through the mesh itself. VMNs are portable, quiet, easy-to-use, and very efficient devices with shorter treatment times and lower residual volume left in the chamber [36]. As reported by Khoo and colleagues, lung levels of aerosolized pirfenidone with a concentration of 2.5 mg/mL—total dose 100 mg—delivered with eFlow Nebulizers, a VMN, were on average 35-fold higher compared to that reported with an oral administration of the approval dose. More specifically, for 100 mg of inhaled pirfenidone, the authors found an average epithelial lining fluid (ELF) Cmax of 135.9 ± 108 ug/mL in comparison with 3.9 ug/ml following an oral dose of 801 mg. Data appear particularly remarkable when it is considered that the efficacy of pirfenidone is related to peak ELF concentration rather than the ELF area under the curve (AUC). Interestingly, the peak plasma mean concentration after a 100 mg nebulized dose is 1.7 ug/ml, a value that is markedly reduced when compared with a median standard plasma mean concentration of 7.9 uL/mL reached following an 801 mg oral dose [37].

#### 2.1.6. Novel Inhalation Formulations: Nanoparticles

The application of nanomedicine, which refers to the use of nanoscale particles as either active molecules or drug delivery vehicles in pharmaceutical formulations, to the treatment of ILDs has been investigated recently. Potentially delivered with DPIs and pMDIs as well as nebulizers, nanoparticles (NPs) may overcome some established aerosol therapy drawbacks. In this respect, the use of NPs may allow for the achievement of higher drug lung deposition, and lower macrophage-mediated clearance [38]. These systems have helped to improve the pharmacokinetics and biodistribution of drugs, reduce side effects, and provide targeted delivery, but inhaled nano-based drug delivery systems still have certain limitations due to specific challenges in construction [39,40]. As drug delivery systems, the carrier materials mainly consist of polymers, lipid-based nanocarriers, or microspheres, and they have different building elements and internal structures [41,42]. Polymeric systems typically exhibit high nebulization stability with minimal toxicity to the lung but have reduced retention, as they often penetrate more easily into the systemic circulation due to their small size [43,44]. Lipid-based systems closely resemble the surfactant lining of the lungs and thus have prolonged retention and better safety in the lungs; however, some systems have shown instability during nebulization [45,46]. Various nanoparticle strategies have been studied and developed for the treatment of lung fibrosis; for example, Wang et al. used polymers, in particular poly (lactic-co-glycolic acid), as a delivery system for nintedanib [47], while Han et al. prepared lipid nanoparticles (Lip@VP) loaded with verteporfin (VER) and pirfenidone [48]. Lipid-based systems in fact have the ability to load multiple types of drugs in a single system.

## 3. Pharmacological Inhaled Agents with Antifibrotic Properties

### 3.1. Nintedanib

Nintedanib is an indolinone derivative that was originally designed as an anti-angiogenic drug targeting the receptor tyrosine kinases VEGFR, FGFR, and PDGFR for the treatment of cancer. Its approval as an antifibrotic treatment in both IPF and PF-ILDs relies on its capability to interfere with lung fibroblast proliferation and migration as well as myofibroblast transformation [49] (Figure 1). Oral nintedanib reaches the pulmonary compartment, but the lung dose may limit access to the airway, alveolar epithelial surface, and possibly underlying fibroblastic foci, the therapeutically relevant target in IPF lungs [50]. Thus, a very large and frequent dose of oral nintedanib is required to achieve a therapeutic effect in the lung; in fact, only the highest oral clinical dose was shown to be effective (150 mg twice daily) [6]. The efficacy of inhaled nintedanib has been investigated recently [50,51] (Table 1). In one of the studies, inhaled nintedanib was able to reach oral-equivalent lung concentrations with significantly lower systemic exposure. Moreover, inhaled nintedanib also improved weight gain in bleomycin-challenged animals, unlike oral administration [51]. To optimize the efficacy of inhaled formulation, nintedanib was loaded into inhalable poly (lactic-co-glycolic acid) (PLGA) nanoparticles (Nint NPs) and tested on A549, primary airway epithelial cell (AEC), and normal human lung fibroblast (NHLF) cell lines. Nint NPs displayed superior antifibrotic activity by inhibiting transforming growth factor beta (TGF-β) signaling, epithelial–mesenchymal transition (EMT), fibroblast to myofibroblast differentiation through the suppression of α-SMA expression, and expression of inflammatory cytokine IL-17A. In addition, Nint NPs increased autophagy, which probably favored collagen resolution in the lung tissues [47]. As mentioned above, nintedanib has been further licensed for the treatment of chronic fibrosing interstitial lung diseases with a progressive phenotype, including occupational pneumoconiosis. In a mouse model of silicosis, the therapeutic efficacy of a nanocrystal-based nanosuspension (NS) formulation of nintedanib (NTB-NS), directly administrated into the lung via intratracheal administration, has been evaluated [52]. This formulation of nintedanib had non-adhesive surface coatings and microscopic dimensions to minimize the adhesive interactions with airway mucus and lung-resident macrophages after inhaled administration, thus increasing the drug concentration in the lung. The authors found that intratracheal NTB-NS given at a 0.1 or 1 mg/kg NTB dose, unlike 100 mg/kg oral NTB, significantly reduced the fibrotic score compared to the untreated silicotic animals. Intratracheal NTB-NS given at 1 mg/kg, but not the daily oral dose of NTB at 100 mg/kg, also significantly reduced the area occupied by silicotic granuloma, the expression of TGF-β1 in the lung tissues, and the static lung elastance. In conclusion, intratracheal NTB-NS showed antifibrotic activity in a mouse model of silicosis without adverse events and provided normalization of lung function at a 100-fold lower dose and a 3-fold lower dosing frequency compared to the oral nintedanib group [52].

### 3.2. Pirfenidone

Similarly, inhaled pirfenidone has been tested in early studies (Table 2); in the ATLAS study, a phase 1b trial, the safety and efficacy of inhaled pirfenidone (AP01) at a dose of 50 mg once per day or 100 mg two times per day were assessed in IPF patients who were intolerant, reluctant, or ineligible for oral pirfenidone or nintedanib [55]. The most commonly reported adverse events (AEs) included cough, rash, dyspnea, nausea, and progression or exacerbation of IPF, fatigue, and lower or upper respiratory tract infections. The reduced incidence of systemic adverse effects is caused by the lower systemic exposure of the nebulized pirfenidone in comparison with oral formulations. Following an 801 mg oral dose, systemic absorption is about 85% (680 mg), while a 100 mg nebulizer dose leads to less than 1/15th of the systemic exposure of the oral dose, with about 45% (45 mg) systemic absorption of the nebulizer dose. Moreover, liver toxicity was mild or moderate in almost all patients due to no first-pass effect. The most commonly reported treatment-related adverse event (AE) was cough, which was self-resolving or controllable by salbutamol. At the same time, as with oral pirfenidone, AP01 decreased the frequency of cough in IPF patients with high baseline cough frequency [55,56].

Regarding the efficacy, the 100 mg two times per day dose group showed markedly less reduction in predicted FVC % compared with the 50 mg once per day group and correlated well with changes in quantitative lung fibrosis scores from high-resolution computed tomography (HRCT).

The difference in efficacy between the two doses could be due to different dose frequencies or dose amounts. The pharmacokinetics, safety, and tolerability of 25, 50, and 100 mg doses of aerosolized pirfenidone were also evaluated in another phase 1 study enrolling normal healthy volunteers, smokers, and IPF patients [37]. No serious AEs were reported, and drug-related AEs, such as cough, increased upper airway secretion, dysphonia, headache, and dizziness, were mainly mild.

Alongside a potential reduction in adverse events, inhalation of antifibrotic agents can result in more effective antifibrotic properties possibly slowing the pace of FVC decline [8,12]. The Capacity 004 phase 3 study of oral pirfenidone demonstrated that the reduction in the decline of predicted FVC % was dose-related, so higher lung levels after aerosol administration may also improve efficacy [57]. In the study by Khoo et al., epithelial lining fluid (ELF) pirfenidone concentrations in the bronchoalveolar lavage (BAL) from a 100 mg inhaled dose can range from 35- to 100-fold higher than those obtained with the approved oral dose of 801 mg [37]. A study of aerosolized pirfenidone in a paraquat fibrosis model also showed that the administration of pirfenidone via inhalation achieved similar results to the conventional oral route at substantially lower doses [53]. Pirfenidone in oral and inhalation routes have comparable effects in decreasing oxidative stress and expression of proinflammatory and profibrotic genes including TGF-b1, tumor necrosis factor alpha (TNF-a), tissue inhibitor of metalloproteinase 1 (TIMP-1), and matrix metalloproteinase 2 (MMP-2) genes. Finally, recent research focused on a dual inhaled drug (verteporfin and pirfenidone)-loaded nanoparticle (Lip@VP) with the function of inhibiting airway epithelium fluidization to the alveoli region and fibroblast overactivation [48]. The results of this study demonstrated that Lip@VP reversed IPF as indicated in improved pulmonary function and pathological features, by preventing honeycomb cyst and interstitium remodeling [48].

### 3.3. Treprostinil

Treprostinil, a prostacyclin analog, may be considered in patients with pulmonary hypertension associated with interstitial lung disease (PH-ILD) based on the findings from the INCREASE study [58,59]. The primary endpoint of the INCREASE study was the exercise capacity, assessed by a 6-min walk test. A post hoc analysis of the INCREASE trial showed that inhaled treprostinil was associated also with an improvement in FVC, particularly in patients with IPF [59]. The reason for this FVC improvement could be the antifibrotic property of treprostinil. In vitro and in vivo studies suggest that treprostinil exerts its antifibrotic effect through multiple pathways, such as the activation of prostaglandin E receptor 2 (EP2), the prostaglandin D receptor 1 (DP1), and peroxisome proliferator-activated receptors (PPARs), leading to the inhibition of fibroblast proliferation, collagen overproduction, and fibroblast-to-myofibroblast transition. Treprostinil can also prevent platelet-derived growth factor (PDGF)- and transforming growth factor beta 1 (TGF b1)-mediated profibrotic effects. The inhibition of both TGFb1 and PDGF is singular for treprostinil as other approved therapies for ILDs only suppressed either TGFb1 or PDGF [60]. A post hoc analysis compared changes in the FVC in response to inhaled treprostinil in patients with PAH from the TRIUMPH study and patients with PH-ILD from the INCREASE study [59,61]. This post hoc analysis found that improvements in FVC were seen in patients with PH-ILD but not in those with PAH, supporting the antifibrotic effect of treprostinil [62]. The phase 3 trial (NCT04708782) evaluating the efficacy of inhaled treprostinil in patients with IPF is ongoing; results are expected by 2025.

**Table 2 pharmaceutics-16-01391-t002:** Pharmacological inhaled agents with antifibrotic properties: in vivo studies.

Author	Year	Study Type	Inhalatory Agent	Device	ILD	N. of Patients	Main Outcome/s and Findings
West et al. [55,56]	2023	Randomized, parallel-group, open-label trial	Pirfenidone	PARI investigational eFlow nebuliser	IPF	91	Aerosol pirfenidone was well tolerated. Regarding efficacy, the predicted FVC % remained stable in the 100 mg two times a day dose group.
Khoo et al. [37]	2020	RCT	Pirfenidone	PARI eFlow nebuliser	IPF	44	Administration of pirfenidone via inhalation delivered higher lung levels than that obtained with administration of the oral dose.
Nathan et al. [59,61]	2021	Post hoc analysis of RCT	Treprostinil	Tyvaso^®^ Ultrasonic Nebulizer	PH-ILD	326	Inhaled treprostinil was associated with improvements in FVC, particularly in patients with IPF.
Hirani et al. [63]	2021	RCT	Inhibitor of galectin-3 (TD139)	DPI	IPF	60	Inhaled TD139 was well tolerated and decreased plasma biomarkers associated with IPF progression.

Abbreviations: DPI: dry powder inhaler; FVC: forced vital capacity; ILD: interstitial lung disease; IPF: idiopathic pulmonary fibrosis; NA: not available; NPs: nanoparticles; NTB: nintedanib; PH: pulmonary hypertension; PI3K: phosphoinositide 3-kinase; RCT: randomized control trial.

### 3.4. Novel Inhaled Antifibrotic Molecules Under Investigation

In a randomized controlled, a phase I/IIa dose-ascending trial evaluated the efficacy of inhalational TD139 via DPIs (Plastiape/Berry Bramlage, Lohne, Germany), with a small molecule inhibiting Gal-3, a member of the β-galactoside-binding lectin family, which regulates fibrotic processes and is overexpressed in the BAL fluid of patients with IPF. Sixty participants were recruited, 24 of whom were diagnosed with IPF [63]. TD139 was well tolerated by both healthy and IPF patients: taste disturbance (36.1%) and cough (11.1%) were the most common adverse effects [63]. TRK250 is another molecule capable of reducing the expression of TGFβ1 and collagen production, inhibiting the transcription of TGFβ1 by producing silencing RNA (siRNA) targeting TGFβ1 messenger RNA (mRNA). The results of the phase I, placebo-controlled, double-blind, randomized study assessing the safety and tolerability of single and multiple inhaled doses of TRK250 in subjects with IPF for 4 weeks are yet to be released (Table 2) [64].

The nuclear factor erythroid 2-related factor 2 (Nrf2) represents a transcription factor with a key role in the expression of cytoprotective as well as metabolic genes involved in internal cellular homeostasis, redox balance, and the response to diverse stresses. Nrf2 is normally anchored in the cytoplasm by binding to the E3 ubiquitin ligase Kelch-like ECH-associated protein 1 (Keap1) or phosphorylated by glycogen synthase kinase 3β (GSK-3β), targeting Nrf2 for its ubiquitination and proteasomal degradation. As several diseases are related to oxidative stress and inflammation, it has been demonstrated that Nrf2 may play a protective role in many conditions, including cardiovascular diseases, neurodegenerative diseases, osteoporosis, and lung fibrosis. Inhalation of dimethyl fumarate (DMF), a first-generation Nrf2 activator, was shown to help maintain the balance of superoxide dismutase (SOD) as well as ROS through the activation of the Nrf2/HO-1 signaling pathway with a positive impact in reducing the number of M2 macrophages and the secretion of TGF-β [65].

The phosphatidylinositol 3-kinase (PI3K)/protein kinase B (PKB/AKT) signaling pathway represents one of the key-signaling molecular pathways in pulmonary fibrosis progression. In a preclinical study, an inhaled prodrug pan-PI3K inhibitor—CL27c—was tested in murine models of asthma, severe asthma, and pulmonary fibrosis. The authors found a significant reduction in fibrotic collagen deposition advantages in lung function and reduced mortality while adverse events were negligible [54].

## 4. Pharmacological Inhaled Agents: Beyond Antifibrotic Properties

### 4.1. Anti-Inflammatory Agents

Corticosteroids are widely used in ILDs including idiopathic non-specific interstitial lung pneumonia (iNSIP), hypersensitivity pneumonitis (HP), sarcoidosis, and eosinophilic pneumonia; conversely, there is a strong recommendation against their use in IPF (Table 3). The inhaled route for the administration of corticosteroids may be attractive as this route limits systemic dose and consequently the deleterious side effects, such as weight gain, fluid retention, hyperglycemia, hypertension, osteoporosis, cataracts, and glaucoma. Several research studies have been conducted in different ILDs.

Sarcoidosis is a systemic granulomatous disease that involves the lungs in more than 90% of cases. Accumulation of non-necrotizing granuloma within the lungs may eventually lead to pulmonary fibrosis. Although a dysregulated interplay between immune cells and mediators represents the pathogenetic hallmark of sarcoidosis [66], data on the impact of inhaled corticosteroids (ICSs) on pulmonary sarcoidosis are contrasting, and studies with different molecules and dosages have been published [67]. In 1994 and 1995, Milman and Alberts, respectively, tested budesonide (BUD) at 1200 mcg once daily, while Erkkila tested BUD at 800 mcg twice daily, in 1988 [68,69,70]. Baughman (2002) and du Bois (1999) compared fluticasone (FP) with a placebo at 880 mg and 2000 mg per day, respectively [71,72], and Ludwig, in 2005, compared beclomethasone diproprionate (BDP) extrafine aerosol 800mcg per day with a placebo [73]. Alberts and co-workers found that patients in the budesonide group showed significant improvements in symptom scores and inspiratory vital capacity (IVC), but not in serum angiotensin-converting enzyme (ACE) concentrations, transfer factors of the lungs for carbon monoxide (TLCOs), or chest radiography findings [68]. On the contrary, Erkkila and al. observed no significant changes in lung function tests and chest radiography in the budesonide-treated patients, while a decrease in serum ACE was observed [70]. Milman and du Bois noted no significant difference regarding symptoms or pulmonary function tests between active treatment and placebo groups [69,72]. Baughman also evaluated FP versus placebo as an oral corticosteroid (OCS)-sparing agent and the authors found that there was no statistically significant difference between FP and the placebo [71]. Ludwing et al. suggested that in patients with pulmonary sarcoidosis and minor functional deterioration, inhalation of high doses of BDP is associated with a decrease in the percentage of bronchoalveolar lavage (BAL) lymphocytes, in parallel with improvements in chest radiography and diffusing capacity for carbon monoxide (DLCO) [73]. Also, Spiteri et al. found a significant decrease in lavage lymphocytosis and symptomatic relief after inhaled budesonide use, at 800 micrograms twice daily [74].

Pietinalho et al. evaluated the efficacy of oral prednisolone followed by inhaled budesonide in patients with newly diagnosed pulmonary sarcoidosis stages I–III [75]. The authors suggested that sequential oral and inhaled corticosteroid therapy may be effective in patients with stage II disease, leading to a statistically significant effect on DLCO values and serum ACE activity.

Other studies suggested some benefits of ICSs, but most of these were not blinded or controlled, and small numbers of patients were included. For example, Selroos found improvements in symptoms, radiographic infiltrations, and FVC in patients with pulmonary sarcoidosis stages II–III treated with inhaled budesonide but no placebo group was included [76]. Also, Gupta conducted an open study including 113 sarcoidosis patients. The authors did not demonstrate significant advancements, except for clinical symptoms, such as cough and dyspnea [77].

Therefore, the selection of patients who might experience higher benefits from ICSs is not straightforward [78]; previous studies suggest that bronchial hyperresponsiveness may be observed in up to 30% of patients with pulmonary sarcoidosis as well as the presence of marked symptoms, so these preserve pulmonary function tests may be selected for patients who may have higher benefits from using ICSs [79,80].

Another disease particularly sensitive to systemic corticosteroid treatment is eosinophilic pneumonia (EP) [81]. The addition of ICSs could be useful to allow for a tapering or discontinuation of long-term oral corticosteroid therapy in patients with chronic eosinophilic pneumonia (CEP) [82,83]. Although CEP is essentially an alveolar interstitial disease, the involvement of central airways has been hypothesized by the findings on bronchial biopsy and the presence of airflow obstruction, even in the absence of asthma [84,85]. Lavandier and Carrè described a patient with CEP who experienced relapses when the dose of oral prednisolone was reduced. High-dose inhaled therapy BDP (1500 mg once daily) was added and progressively reduced to 1 mg/day. The addition of a high dose of BDP, however, permitted the patient to taper the dose and discontinue OCS use. Later, the patient experienced three acute exacerbations, treated with an increase in high-dose inhaled steroids or a 5-day course of oral prednisone [82]. More recently, Minakuchi et al. evaluated whether ICS monotherapy is effective in four patients with CEP. All patients had worsening or relapse of CEP during the treatment with BDP, so ICSs alone may not be effective in these patients [81]. Placebo-controlled trials in a large number of patients are needed to verify the response to treatment.

**Table 3 pharmaceutics-16-01391-t003:** Inhaled agents with anti-inflammatory properties in ILDs.

Author	Year	Study Type	Inhalatory Agent	Device	Interstitial Lung Disease	N. of Patients	Main Outcome/s and Findings
Alberts et al. [68]	1995	RCT	Budesonide	Nebuhaler	Sarcoidosis stages I–III	47	Inhaled budesonide improved the severity of symptoms and lung function, in particular, IVC.
Milman et al. [69]	1995	RCT	Budesonide	Nebuhaler	Sarcoidosis stages I–III	21	Inhaled budesonide had no evident therapeutic effect.
Erkkila et al. [70]	1988	RCT	Budesonide	NA	Newly diagnosed sarcoidosis	19	Inhaled budesonide induced no significant changes in chest radiography or lung function tests but influenced biochemical and cellular findings.
Baughman et al. [71]	2002	RCT	Fluticasone Propionate	NA	Sarcoidosis	21	There was no significant difference in the improvement of lung function, cough, or average daily dose of prednisone for the fluticasone versus placebo.
Du Bois et al. [72]	1999	RCT	Fluticasone Propionate	NA	Stable pulmonary sarcoidosis	44	Inhaled fluticasone propionate did not offer an objective benefit in stable pulmonary sarcoidosis.
Ludwing et al. [73]	2005	RCT	BDP	Extrafine, HFA	Newly diagnosed pulmonary sarcoidosis stages I–III	15	In patients with pulmonary sarcoidosis of stage II and minor functional impairment, inhalation of BDP is associated with a reduction in the number of BAL lymphocytes, in parallel with improvements in chest radiography and DLCO.
Spiteri et al. [74]	1989	RCT	Budesonide	Nebuhaler	Symptomatic sarcoidosis	25	Inhaled budesonide can induce a significant decrease in lavage lymphocytosis with concomitant symptomatic relief.
Pietinalho et al. [75]	1999	RCT	Budesonide	DPI	Newly diagnosed stage I and stage II pulmonary sarcoidosis	189	An initial treatment with prednisolone followed by long-term inhalation of budesonide could be efficacious in patients with stage II disease, leading to functional improvement and a decrease in SACE activity.
Selroos et al. [76]	1986	Open and uncontrolled clinical study	Budesonide	MDI	Sarcoidosis stages II–III	20	With budesonide, SACE activity was normalized in all patients, and a general improvement in chest radiography and FVC was noted.
Minakuchi et al. [81]	2003	Open and uncontrolled clinical study	BDP	MDI	CEP	4	Monotherapy with BDP may not be effective.

Abbreviations: BAL: bronchoalveolar lavage; BDP: beclomethasone dipropionate; CEP: chronic eosinophilic pneumonia; DLCO: diffusing capacity for carbon monoxide; DPI: dry powder inhaler; FVC: forced vital capacity; IVC: inspiratory vital capacity; MDI: metered-dose inhaler; NA: not available; RCT: randomized controlled trial; SACE: serum angiotensin-converting enzyme.

### 4.2. Mucolytics and Antioxidant Agents

Originally, N-acetylcysteine (NAC) was routinely used in patients with IPF. However, in 2005, the PANTHER study was prematurely discontinued based on excess mortality outcomes in the arm of patients treated with the combination of prednisone, azathioprine, and NAC. However, the pharmacological role of NAC in ILDs was further subsequently explored. NAC could express pharmacological interest in fibrotic interstitial lung diseases as it may interfere with many proinflammatory pathways. In particular, in two experimental models S-allylmercapto-NAC and a combination of NAC with desipramine—an antidepressive agent inhibiting the acid sphingomyelinase (ASMase)/ceramide signaling pathway—counterbalances the profibrotic effects blunting Nrf2/HO-1 signaling and inhibits key regulator inflammatory/fibrotic pathways such as the NF-κB and TGF-β1/Smad2/3 pathways [86,87]. Clinical data testing the addition of NAC to the antifibrotic agents are controversial. In 2015 and 2016, two independent research studies documented that the inhalation of NAC may influence the FVC decline in IPF patients [88,89]. These results prompted further investigations; unfortunately, a subsequent meta-analysis of six studies suggested no evidence of 24-week FVC decline in IPF patients who were treated with NAC in addition to pirfenidone compared with pirfenidone alone. However, in only one study included in the analysis, NAC was administered through an inhalation route [88]. Therefore, in 2021, Sakamoto et al. published a phase III prospective trial testing the addition of inhaled NAC 352.4 mg twice daily with oral pirfenidone; the results documented a worse FVC decline (−300 mL) during the 48-week follow-up in patients treated with the combination of inhaled NAC and oral pirfenidone compared with oral pirfenidone alone [90] (Table 4). However, at this time, there are no data available about the combination of inhaled pirfenidone and NAC. Conversely, the idea of a multitarget intervention in ILDs has led to the development of the co-suspension of NAC and nintedanib. In an experimental murine model, inhaled nintedanib [91] was tested in combination with NAC; Zhang et al. found that the co-loaded liposomes of S-allylmercapto-NAC and nintedanib were more effective than the single drugs in preventing collagen I, collagen III, and α-SMA deposition.

Another mucolytic with antioxidant properties, ambroxol hydrochloride (AH)—the active metabolite of the bromexine—has been investigated in a preclinical model of pulmonary fibrosis in combination with baicalin (BA), an extract from Scutellaria baicalensis with potential antifibrotic effects exhibited through the upregulation of the adenosine A2a receptor, downregulation of TGF-β1, and phosphorylation of ERK1/2. AH and BA were administered via a dry powdered inhaler (DPI). In bleomycin-induced rat models, the co-administration of AH/BA via DPIs prevents fibrotic changes and reduces the expression of inflammatory and fibrotic mediators in the BAL fluid (MPO, IL-4, IL-6, IL-8, IL-1β, and TGF-β1) [92].

Antioxidants are a wide class of drugs with a scavenger effect counteracting the increased oxidative stress and mitochondrial dysfunction observed in ILDs (Table 5). While oral administration is limited due to low local bioavailability, a growing body of research is investigating novel formulations with recent advances in preclinical models.

Resveratrol—a natural non-flavonoid polyphenolic compound—downregulates vimentin and upregulates E-cadherin expression and also blunts TLR4/NF-κB and TGF-β1/smad3 signaling pathways. Therefore, in 2023, Ahmed et al. developed resveratrol-loaded spray-dried composite microparticles delivered via DPIs; the authors found that animals treated with this agent had lower hydroxyproline, tumor necrosis factor-α, and matrix metalloproteinase-9 levels as well fewer fibrotic changes [93]. Likewise, luteolin (3′,4′,5,7-tetrahydroxyflavone), a flavone with anti-inflammatory and antioxidant properties, has been shown to effectively reduce fibrosis progression when loaded into γ-cyclodextrin metal–organic frameworks (LUT@CDMOFs) and administered via DPIs, in a murine model of bleomycin-induced lung fibrosis [94].

Finally, a combination of hydrogen (H2)—a colorless, odorless, gas—and tetrandrine, a natural compound originated from Stephania tetrandrine with potential antioxidative stress, anti-inflammatory, and antifibrotic effects delivered at a rate of 3 L/min via nasal cannula using the hydrogen/oxygen generator, was tested in a rat model of silica. The results documented that the combination of H2 and tetrandrine attenuates the NF-κB/NLRP3 signaling pathway resulting in alleviated oxidative stress, reduced expression of inflammatory biomarkers, and inhibition of the epithelial–mesenchymal transition [95].

### 4.3. Bronchodilators

Bronchodilators are well-established therapeutic options for airway diseases, such as asthma and chronic obstructive pulmonary disease (COPD), improving both breathless and lung function; their use alone or in combination with ICSs in ILDs is summarized in Table 6. Traditionally, IPF has been considered a disease that destroys the alveoli but spares the smaller airways. As early as 1970, this hypothesis was rejected by Mead, supporting the hypothesis that IPF is also characterized by small airway disease [96]; then, in 2020, via a comprehensive analysis of IPF lung explants using a cascade of clinical multi-detector CT (MDCT) scans, Verleden et al. found that thickening of small airway walls and distortion of small airway lumens could increase the visibility of small airways on MDCT in IPF patients. In addition, compared with normal lung anatomy, the number of terminal bronchioles was reduced within regions of minimal fibrosis in IPF lungs. These data indicate that the pathology in the small airways could be an early feature of IPF, a precursor of the change seen in large airways [97].

In addition, IPF may be comorbid with obstructive lung diseases. Combined pulmonary fibrosis and emphysema (CPFE) is a clinical entity characterized by the combination of upper lobe emphysema and lower lobe fibrosis. Among the ILDs with concomitant emphysema, IPF is the most frequent. Patients with CPFE have limited exercise capacity, severely impaired gas exchange, and relatively preserved airflow rates and lung volumes [7,98,99,100]. In patients with normal or subnormal spirometry, CPFE may be under-recognized if DLCO is not measured [98]. In CPFE patients, the use of inhaled bronchodilators is debated due to the relatively well-preserved values; the data are discordant and there is a lack of general consensus on the specific treatment of emphysema in the setting of IPF. The French guidelines for the management of IPF suggest that inhaled bronchodilators should be used if airflow obstruction is present in patients with IPF and emphysema [101], and the Spanish guidelines for the treatment of IPF propose that for patients with obstructive or mixed functional defects, inhaled bronchodilators may be prescribed [102]; however, most patients with IPF have a forced expiratory volume in one second (FEV1)/FVC ratio >0.8.

Dong et al. assessed the efficacy of the ICS/long-acting β2-agonist (LABA) combination in patients with CPFE [103]. The authors showed that ICS/LABA therapy could improve lung function in these patients; in particular, FEV1 %, FVC %, and DLCO% significantly improved in the ICS/LABA treatment group after 12 months, and lung HRCT scores significantly declined in comparison with the non-treatment group. Moreover, ICS/LABA could significantly decrease the frequency of acute exacerbations. Zhang et al. evaluated the effect of different therapeutic regimens on the prognosis of CPFE patients, assessed by the composite physiologic index (CPI) and high-resolution computed tomography (HRCT) scores. The authors found that CPFE patients treated with oxygen therapy, bronchodilators, and corticosteroids had a lower rate of increase in CPI than patients in the other treatment groups [104].

Hu et al. showed that it is not emphysema but functional parameters of small airways that are a deciding factor for guiding bronchodilator therapy in IPF patients [105]. The authors detected small airway dysfunction (SAD), a pathological condition that affects the bronchioles and non-cartilaginous airways 2 mm or less in diameter, in patients with IPF using impulse oscillometry (IOS). IPF patients with SAD, defined according to IOS parameters, had a significant improvement in FEV1, forced expiratory fraction at 25–75% of FVC (FEF25%-75%), and symptom score after bronchodilator treatment. Bronchodilator efficacy was not observed in IPF patients without SAD or with emphysema. In conclusion, small airways may represent a neglected therapeutic target in IPF to improve symptoms and slow disease progression, but further investigations are needed to understand the role of the small conducting airways in these patients.

Also, Zhang et al. concluded that SAD might help manage IPF patients; furthermore, the authors found that SAD worsens the prognosis of IPF patients [106].

There could be a pharmacological rationale supporting LABA use in IPF as human lung fibroblasts express β2-adrenoceptors and agonist-induced activation may suppress the profibrotic activity of lung fibroblasts [107]. Epidemiological evidence suggests a propensity for platelet activation and thrombosis in IPF patients [108], so Wright et al. demonstrated that “ultrafine” inhaled beclomethasone/formoterol significantly reduced platelet activation, which consequently induced proinflammatory and profibrotic mediator release. In addition, this combination was associated with improvement in FEV1 and FEF 25-75% compared to the placebo [109]. Also, Herrmann et al. assessed the antifibrotic activity of a LABA, Olodaterol, in primary human lung fibroblasts from control donors (HLFs) and patients with IPF (IPF-LFs) [110]. Their data demonstrated that in vitro Olodateral blocked transforming growth factor (TGF)-β-induced myofibroblast differentiation and the fibroblast growth factor (FGF)- and platelet-derived growth factor (PDGF)-induced migration and proliferation of cells in both HLFs and IPF-LFs. In addition, in a murine model of lung fibrosis, Olodaterol not only inhibited the release of profibrotic mediators, but preventive treatment also attenuated the decline in FVC.

FVC is a key measure of disease severity in patients with interstitial lung diseases, and according to the latest clinical guidelines, an absolute decline in FVC of >5% within 1 year of follow-up has been considered as criteria for the definition of progressive pulmonary fibrosis [111]. IPF may often have comorbid obstructive airway disease with some degree of reversibility that could influence FVC measurement; the use of a bronchodilator prior to FVC measurement in patients with IPF may reduce the risk of misclassification of disease progression due to the variable presence of reversible airflow limitation in a minority of IPF patients [112]. Hypersensitivity pneumonitis (HP) is one of the most frequent causes of distal airway disease with small airway abnormalities characterized by lymphocytic infiltrates and granuloma formation, leading to bronchial obstruction [113]. A study by Dias et al. evaluated the small airway involvement in fibrotic HP patients with restrictive ventilatory patterns; the authors demonstrated that no patient exhibited a bronchodilator response using a forced oscillation technique (FOT), detected as a fall > 40% of the resistance at 5 Hz (R5) values or a significant difference in resistance at 5HZ of resistance at 19 Hz (R5–19) values after using a bronchodilator, although R5 values upon inspiration were significantly lower after salbutamol use. The role of bronchodilators in obstructive HP patients remains to be evaluated because a small percentage of patients with fibrotic HP present obstruction in pulmonary functional tests [114].

Sarcoidosis can affect the airway at any level, from the upper respiratory tract to small airways. Although pulmonary sarcoidosis characteristically causes a restrictive ventilatory defect seen in pulmonary functional tests, an obstructive pattern seems to be very common. Broncho stenosis could be due to chronic granulomatous inflammation of the bronchial wall, endobronchial masses, extrinsic compression from enlarged lymph nodes, or small airway distortion [115]. In a cohort of 15 patients with chronic pulmonary sarcoidosis and airway obstruction, Young et al. demonstrated poor reversibility after inhalation of isoproterenol [116].

**Table 6 pharmaceutics-16-01391-t006:** Inhalation agents with bronchodilator properties in ILDs.

Author	Year	Study Type	Inhalatory Agent	Device	Interstitial Lung Disease	N. of Patients	Main Outcome/s and Findings
Dong et al. [103]	2015	Prospective cohort study	ICS/ LABA	NA	CPFE	103	Lung function and radiological scores significantly improved in the ICS/LABA treatment group in comparison to the non-treatment group.
Zhang et al. [104]	2016	Retrospective cohort study	Bronchodilators	NA	CPFE and IPF	192	The rate of increase in CPI score in the group treated with oxygen therapy, bronchodilators, and OCS was lower than that in the other groups.
Hu et al. [105]	2020	Retrospective cohort study	LAMA, LAMA/LABA, ICS/LABA, and LAMA/LABA/ICS	NA	IPF	63	In IPF patients with SAD, there was significant improvement in lung function after bronchodilator treatment. In IPF patients with emphysema, there were no significant differences after bronchodilators.
Wright et al. [109]	2017	RCT	Beclomethasone/ Formoterol	pMDI	IPF	17	Beclomethasone/formoterol significantly reduced platelet reactivity and improved FEV1 and FEF25–75%.
Dias et al. [114]	2019	Prospective cohort study	Bronchodilators	NA	HP	28	No patient exhibited a bronchodilator response detected at FOT.
Young et al. [116]	1980	Retrospective cohort study	Isoproterenol	NA	Pulmonary sarcoidosis	15	Patients with airway obstruction had poor reversibility after inhalation of isoproterenol.

Abbreviations: CPFE: combined idiopathic pulmonary fibrosis and emphysema; CPI: composite physiologic index; ICSs: inhaled corticosteroids; FEF25–75%: forced expiratory fraction at 25–75% of FVC; FEV1: forced expiratory volume in 1 s; FOT: forced oscillation technique; HP: hypersensitivity pneumonitis; IPF: idiopathic pulmonary fibrosis LABA: long-acting beta2 agonist; LAMA: long-acting muscarinic antagonist; pMDI: pressurized metered-dose inhaler; OCSs: oral corticosteroids; RCT: randomized controlled trial; SAD: small airway dysfunction.

## 5. Future Perspectives and Conclusions

Remarkable progress has been achieved in the treatment of ILDs. Data on the efficacy and safety of inhaled treprostinil in ILD-related pulmonary hypertension have been published leading to its approval. However, a consistent number of patients, ranging from 8 to 48% [13,117], experience serious medication-related adverse effects leading to temporary as well as permanent treatment stoppages. In this respect, ameliorating the tolerability of current medications as well as identifying new targetable pathways represent a real challenge for clinicians. Directly delivered into the lungs, an inhaled medication may limit fibrogenetic processes, minimizing systemic side effects. Investigation of novel molecules and formulations is currently ongoing in new recruiting trials (Table 7).

## Figures and Tables

**Figure 1 pharmaceutics-16-01391-f001:**
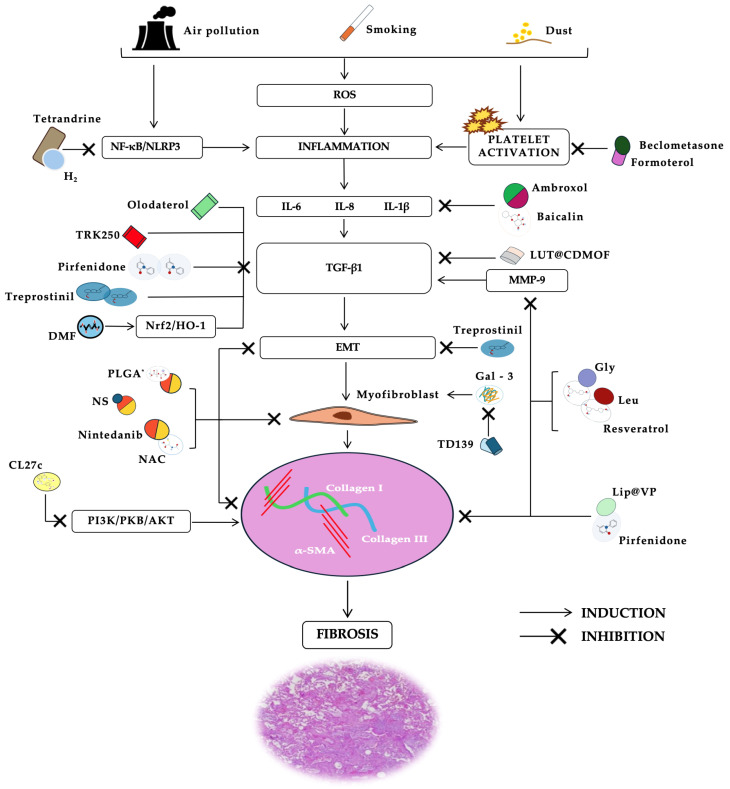
Inhaling molecules under investigation in pulmonary fibrosis treatment. Smoking, air pollution, and dust represent the main noxae involved in ROS production and inflammation ignited within the lungs. This eventually may lead to a sustained aberrant response of alveolar epithelial cells leading EMT to progress and, finally, to fibrosis and lung parenchyma destruction. Several inhaling molecules are currently under investigation for future employment in pulmonary fibrosis treatment. *: PLGA–nintedanib has currently only been tested in vitro, by Wang et al. [36]. α-SMA: alpha-smooth muscle actin; DMF: dimethyl fumarate; Gal-3: galectin 3; Gly: glycine; EMT: epithelial–mesenchymal transition; HO-1: heme oxygenase 1; Leu: leucine; Lip@VP: Liposome@verteporfin/pirfenidone; LUT@CDMOFs: Luteolin into γ-cyclodextrin metal–organic frameworks; MMP-9: matrix metalloproteinase-9; NAC: N-acetyl cysteine; NF-κB: nuclear factor kappa B; NLRP3: NLR family pyrin domain-containing 3; Nrf2: nuclear factor erythroid factor 2-related factor 2; NS: nanosuspension; PI3Ks: phosphoinositide 3-kinases; PKB: protein kinase B; PLGA: poly lactic-co-glycolic acid; ROS: reactive oxygen species; TD139: thiodigalactoside galectin-3 inhibitor; TGF-β1: transforming growth factor beta 1.

**Table 1 pharmaceutics-16-01391-t001:** Pharmacological inhaled agents with antifibrotic properties: in vitro and murine models.

Author	Year	Study Type	Inhalatory Agent	Device	ILD	N. of Patients	Main Outcome/s and Findings
Surber et al. [51]	2020	Murine model	Nintedanib	Aerosolizer^®^	Bleomycin-induced pulmonary fibrosis	NA	Inhalation delivers superior nintedanib lung C-max with substantially lower systemic exposure compared to oral administration.
Epstein-Shochet et al. [50]	2020	Murine model	Nintedanib	NA	Silica-induced pulmonary fibrosis	NA	Inhaled nintedanib pharmacokinetics are effective for treating silica-induced pulmonary fibrosis and reducing pulmonary inflammatory markers in mice.
Rasooli et al. [53]	2018	Murine model	Pirfenidone	Ultrasonic nebulizer	Paraquat-induced lung fibrosis	NA	Pirfenidone in oral and inhalation routes has similar therapeutic effects; however, the dose of the inhalation route was lower.
Han et al. [48]	2024	Murine model	Lipid nanoparticles (Lip@VP) loaded with verteporfin and pirfenidone		Lung fibrosis	NA	Lip@VP inhibited the formation of honeycomb cysts and interstitium remodeling.
Wang et al. [47]	2024	In vitro	Nintedanib nanoparticles (NTB NPs)	NA	IPF	NA	NTB NPs counteracted the epithelial-to-mesenchymal transition by limiting collagen production and deposition and fibroblast-to-myofibroblast differentiation. Cultured IPF cells treated with NTB-NPs expressed elevated E-cadherin levels and reduced and decreased the expression of IL-17A.
Da Silva et al. [52]	2023	Murine model	Nanocrystal-based suspension formulation of nintedanib (NTB-NS)	Intratracheal administration	Silicosis	NA	NTB-NS produced significant antifibrotic effects and mechanical lung functional recovery at a 100-fold lower dose and a 3-fold lower dosing frequency compared to oral NTB.
Campa et al. [54]	2018	Murine model	Prodrug pan-PI3K inhibitor (CL27c)	NR	Lung fibrosis	NA	CL27c was able to reduce lung damage and prevent animal mortality.

Abbreviations: IPF: idiopathic pulmonary fibrosis; NA: not applicable; NR: not reported; NTB: nintedanib.

**Table 4 pharmaceutics-16-01391-t004:** NAC in ILDs: in vivo studies.

Author	Year	Study Type	Inhalatory Agent	Device	ILD	N. of Patients	Main Outcome/s and Findings
Sakamoto et al. [88]	2015	Case–control study	Inhaled NAC	Nebulizer	IPF	34	Combination treatment with inhaled NAC and oral pirfenidone reduced the rate of annual FVC decline and improved PFS.
Okuda et al. [89]	2016	Prospective study	Inhaled NAC	Nebulizer	IPF	28	Inhaled NAC significantly reduced the decline in FVC in patients with mild to moderate IPF, particularly in those with progressive disease.
Sakamoto et al. [90]	2021	Phase 3, randomized, open-label study	Inhaled NAC	Nebulizer	IPF	81	The decline in FVC was greater in patients treated with pirfenidone plus inhaled NAC than in those treated with pirfenidone alone.

Abbreviations: RCT: randomized controlled trial; IVC: inspiratory vital capacity; NA: not available; BDP: beclomethasone dipropionate; BAL: bronchoalveolar lavage; DLCO: diffusing capacity for carbon monoxide; DPI: dry powder inhaler; SACE: serum angiotensin-converting enzyme; MDI: metered-dose inhaler; FVC: forced vital capacity; CEP: chronic eosinophilic pneumonia.

**Table 5 pharmaceutics-16-01391-t005:** Inhaled antioxidant agents in pulmonary fibrosis: murine models.

Author	Year	Study Type	Inhalatory Agent	Device	ILD	Main Outcome/s and Findings
Zhang et al. [91]	2024	Murine model	Inhalable ASSNAC and nintedanib co-loaded liposomes	NA	Bleomycin-induced PF	The combination of ASSNAC and nintedanib reduced the expression of collagen I, collagen III, and α-SMA.
Qi et al. [92]	2023	Murine model	BA/AH	DPI	Bleomycin-induced PF	BA/AH DPIs significantly improved lung function, and exhibited antifibrosis, anti-inflammatory, and antioxidant effects.
Ahmed et al. [93]	2023	Murine model	Resveratrol	DPI	Bleomycin-induced PF	The formulation used alleviated PF by suppressing the levels of hydroxyproline, tumor necrosis factor-α, and matrix metalloproteinase-9.

Abbreviations: ASSNAC: S-allylmercapto-N-acetylcysteine; BA/AH: baicalin/ambroxol hydrochloride; DPI: dry powder inhaler; PF: pulmonary fibrosis.

**Table 7 pharmaceutics-16-01391-t007:** Current inhaled agents under investigation in ILDs: recruiting trials.

Molecule	Disease	Device	Sponsor	Phase/Status	Estimated Study Completion	Study
Treprostinil (TPIP)	PH-ILD	DPI	Insmed Incorporated	OLE Recruting	03/2026	NCT05649722
Treprostinil	PH-S	Tyvaso^®^	University of Florida	Phase II Recruiting	07/2025	NCT03814317 (SAPPHIRE)
Treprostinil	PPF	Ultrasonic Nebulizer Tyvaso^®^	United Therapeutics	Phase III Recruiting	11/2027	NCT05943535 (TETON-PPF)
Treprostinil	IPF	Ultrasonic Nebulizer Tyvaso^®^	United Therapeutics	Phase III Recruiting	06/2025	NCT04708782 (TETON)
Treprostinil (L606)	PH-ILD	Liposomal (L606)	Liquidia Technologies, Inc.	Phase III Recruiting	07/2025	NCT04691154
ARO-MMP7	IPF	Nebulized Solution	Arrowhead Pharmaceuticals	Phase I/IIa Recruiting	03/2025	NCT05537025
Pirfenidone (AP01)	PPF	Oral Inhalation Solution	Avalyn Pharma Inc.	Phase IIb Recruiting	04/2026	NCT06329401
CEFFE	PF	Nebulized Solution	Shanghai Ninth People’s Hospital -Shanghai Jiao Tong University	Phase I Recruiting	12/2024	NCT05883293 (CEFFE-PF)
CMR316	IPF	Nebulized Solution	Calibr Scripps Research	Phase I/Ib Recruiting	03/2026	NCT06589219

Abbreviations: CEFFE: cell-free fat extract; DPI: dry powder inhalation; OLE: open-label extension; TPIP: treprostinil palmitil inhalation powder.

## Data Availability

Not applicable.

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
