# Peer review of "Pharmacological Treatment of Interstitial Lung Diseases: A Novel Landscape for Inhaled Agents"

_pharmaceutics, 2024, doi:10.3390/pharmaceutics16111391_

Round 1

Reviewer 1 Report

Comments and Suggestions for Authors

Pharmacological Treatment of Interstitial Lung Diseases – This is a well-written article.

To further enrich the content, kindly consider the following points:

  1. In the Therapeutic Aerosols section, please include details about soft mist inhalers (SMIs). SMIs are a new technology and are expected to be used for delivering both biological agents and small molecules.

  2. The author has covered most of the repurposed molecules, and the title reflects a Novel Landscape for Inhalatory Agents. However, it would be beneficial if the author reviews the pipelines of various multinational companies to explore current molecules under clinical evaluation.

Comments on the Quality of English Language

NA

Author Response

To further enrich the content, kindly consider the following points:

In the Therapeutic Aerosols section, please include details about soft mist inhalers (SMIs). SMIs are a new technology and are expected to be used for delivering both biological agents and small molecules.

  • We wish to thank the Referee for his/her kind suggestion. We accordingly have added a few lines about SMI technology which may offer several advantages in providing drugs limiting involuntary poor adherence.

The author has covered most of the repurposed molecules, and the title reflects a Novel Landscape for Inhalatory Agents. However, it would be beneficial if the author reviews the pipelines of various multinational companies to explore current molecules under clinical evaluation.

  • We wish to thank the Referee for his/her valuable comment. Accordingly, we have added a new table at the end of the manuscript in the Conclusion and Future Perspective session highlighting main ongoing trials on inhalers and ILD.

Reviewer 2 Report

Comments and Suggestions for Authors

The aim of the paper is the evaluate the promises and the results of inhaled therapy in ILD patients. This argument is very interesting and includes two large fields of research: ILD and aerosol therapy.

ILD that are heterogenoues groups of diseases. Into the introduction section, authors initially seem to limit their research to “Idiopathic pulmonary fibrosis (IPF) …………..and progressive fibrosing ILDs (PF-ILDs).” Differently, in the following parts of the paper, they discuss of sarcoidosis, and pulmonary hypertension.

Into the section 2, authors write “the ideal inhaler should be a convenient, easy-to-use, environmentally sustainable, humidity-resistant device, able to provide the accurate amount of drugs into the lungs, also in different patient’s conditions and capable to provide feedback to patients for each either properly or improperly dose taken “. Effectively, aerosol therapy, a not easy way of treatment. By contrast, authors largely describe principles of aerosol therapy, generics of DPI, MDI, nebulized therapy, that could be substitute with some key references. On the contrary, they do not discuss the inhalers used in experimental studies in ILD subjects.

As a conclusion, authors include a large range of studies and arguments, but the current final reading is hard, in my opinion. Their work is huge, but I suggest that they re-organize the paper, limiting the content of the paper to some key-messages

Authors include in vitro studies, studies on animal and humans, unifying them in single tables (please separate them).

They describe experimental or almont-experimental studies aiming to inhaled treatment of lung fibrosis. In addition, they evaluate the role fo ICS and bronchodilators in ILD subjects. ICS and bronchodilators are a standard of treatment in asthma and COPD, but they are also used in ILD or mixed ILD-COPD. It is interesting to evaluate the evidence of ICS and LABA treatment, often observed in real life, in ILD patients. Again, I think that they put too many arguments into the paper.

They should shorten their work, taking off some generic parts (i.e.

Comments on the Quality of English Language

I'm not qualified to assess the quality of English

Author Response

The aim of the paper is the evaluate the promises and the results of inhaled therapy in ILD patients. This argument is very interesting and includes two large fields of research: ILD and aerosol therapy.

ILD that are heterogenoues groups of diseases. Into the introduction section, authors initially seem to limit their research to “Idiopathic pulmonary fibrosis (IPF) …………..and progressive fibrosing ILDs (PF-ILDs).” Differently, in the following parts of the paper, they discuss of sarcoidosis, and pulmonary hypertension.

-          We wish to thank the Referee for his/her valuable comment. We do agree that pulmonary hypertension (despite being a potential complication of ILDs) is probably out of topic and in the revised version of the manuscript we have removed the paragraph about agents with vasodilator Properties. We have left into the manuscript the role of inhaled Treprostinil who is under investigation for its antifibrotic properties. We also included discussion about sarcoidosis which is actually included among ILDs – despite being a multisystemic granulomatous disorder – according the Travis’s Classification  (https://www.ncbi.nlm.nih.gov/pmc/articles/pmid/24032382/). According to your insightful comment, we have introduced and briefly discuss sarcoidosis in the Paragraph of Anti-Inflammatory Agents (Lines 385-390). Moreover, in the Introduction section, we made explicit that nintedanib has been approved for ILDs, other than IPF, highlighting terms such as sarcoidosis and CTD-ILDs.

Into the section 2, authors write “the ideal inhaler should be a convenient, easy-to-use, environmentally sustainable, humidity-resistant device, able to provide the accurate amount of drugs into the lungs, also in different patient’s conditions and capable to provide feedback to patients for each either properly or improperly dose taken “. Effectively, aerosol therapy, a not easy way of treatment. By contrast, authors largely describe principles of aerosol therapy, generics of DPI, MDI, nebulized therapy, that could be substitute with some key references. On the contrary, they do not discuss the inhalers used in experimental studies in ILD subjects. As a conclusion, authors include a large range of studies and arguments, but the current final reading is hard, in my opinion. Their work is huge, but I suggest that they re-organize the paper, limiting the content of the paper to some key-messages

  • We wish to thank the Referee for his/her valuable comment. Accordingly, we have removed some redundant part about aerosol technique ad we have now included more details about novel inhalers (including SMIs) and technology (i.e. nanoparticles). In addition, we have added a new Table (Table 7) highlighting new ongoing trials on inhalers in ILDs, with new molecules and formulations and a brief section on Soft Mist Inhalers, as suggested by Reviewer 2. We hope you may appreciate these changes

Authors include in vitro studies, studies on animal and humans, unifying them in single tables (please separate them).

  • We wish to thank the Referee for his/her valuable comment. In the new version we have divided the tables accordingly.

They describe experimental or almont-experimental studies aiming to inhaled treatment of lung fibrosis. In addition, they evaluate the role of ICS and bronchodilators in ILD subjects. ICS and bronchodilators are a standard of treatment in asthma and COPD, but they are also used in ILD or mixed ILD-COPD. It is interesting to evaluate the evidence of ICS and LABA treatment, often observed in real life, in ILD patients.

  • We wish to thank the Referee for his/her valuable comment. While we do agree that ICS and bronchodilators (BDs) are often use in clinical practice also in patients with coexisting ILD they are approved for COPD (and asthma). Therefore, we summarized in the paragraph current limitation in clinical practice (in COPD BDs are approved only for patients with FEV/FVC < 5th percentile) while due to presence of coexisting fibrosis and restrictive pattern these agents should be better investigated in this latter group and solid evidence still lack.

Again, I think that they put too many arguments into the paper.

They should shorten their work, taking off some generic parts (i.e.

  • We wish to thank the Referee for his/her valuable comment. In this version some parts have been removed (PH) or shortened (aerosol principles). We thank you for your time and effort in reviewing our manuscript.